# Investigation of Energy Levels of Small Vacancy Clusters in Proton Irradiated Silicon by Laplace Photoinduced Transient Spectroscopy



Paweł Kamiński [1], Jarosław Żelazko [1,*], Roman Kozłowski [1], Christian Hindrichsen [2] and Leif Jensen [2]

1   Łukasiewicz Research Network, Institute of Microelectronics and Photonics, Aleja Lotników 32/46, 02-668 Warszawa, Poland
2   Topsil GlobalWafers A/S, Siliciumvej 1, DK-3600 Frederikssund, Denmark
*   Correspondence: jaroslaw.zelazko@imif.lukasiewicz.gov.pl

**Abstract:** Laplace photoinduced transient spectroscopy has been applied to determine the electronic properties and concentrations of deep traps in high purity *n*-type silicon irradiated with high fluences of 23-MeV protons. From the temperature dependence of thermal emission rates of excess charge carriers obtained by the analysis of the photocurrent relaxation waveforms measured at temperatures of 30–320 K, eight deep traps with activation energies ranging from 255 to 559 meV have been resolved. The dependence of these trap's concentrations on the proton fluence are demonstrated for the fluence values ranging from $1 \times 10^{14}$ to $5 \times 10^{15}$ $n_{eq}/cm^2$. In comparison to the previously reported results of theoretical and experimental studies on the electronic properties of small vacancy clusters in irradiated silicon, we tentatively attribute four detected traps with activation energies of 255, 367, 405, and 512 meV to the energy levels related to the $2-/-$ charge state changes of divacancy ($V_2$), trivacancy ($V_3$), tetravacancy ($V_4$), and pentavacancy ($V_5$), respectively. Simultaneously, we propose the attribution of four deep traps with higher activation energies of 415, 456, 526, and 559 meV to the energy levels related to the $-/0$ charge state changes of these small vacancy clusters, respectively.

**Keywords:** Laplace PITS; energy levels; high-purity silicon; vacancy clusters; radiation defects

## 1. Introduction

Knowledge on the properties of irradiation-induced point defects in crystals is of great importance in terms of the various materials applications in the nuclear power industry, nuclear medical diagnostics, and therapy, as well as in high-energy particle physics. The vast majority of these materials are used in sensors or detectors of particles arising from nuclear reactions and are exposed to nuclear radiation. As a result, radiation induced defects are formed in these materials, leading to the device's performance degradation and the necessity to replace them. Crystalline silicon is most frequently used as a basic material of electron devices working in a nuclear radiation environment, and understanding of the lattice defect properties and formation mechanisms is vital to improve the device's radiation hardness [1–4].

Vacancies are fundamental, electrically active, irradiation-induced point defects in silicon affecting the material's electrical, optical, and mechanical properties. These primary defects are formed under irradiation of silicon with high-energy particles or ions, as well as due to plastic deformation [1–3]. They are very mobile and tend to aggregate, forming clusters. For example, the diffusion of the doubly negatively charged vacancies is characterized by an activation energy of 0.18 eV, and this value gives a diffusion coefficient at 300 K of $1.42 \times 10^{-5}$ $cm^2/s$ [3]. The activation energy for the neutral vacancy's diffusion is 0.45 eV and the corresponding value for the diffusion coefficient at 300 K is $3.6 \times 10^{-11}$ $cm^2/s$ [3].

Small vacancy clusters, $V_n$ ($2 \leq n \leq 5$), are electrically active defects that are of great importance in terms of their damaging effect on the performance of silicon devices, in particular on the characteristics of silicon detectors used to track high energy particles

which are exposed to irradiations with high fluences of fast hadrons. Results obtained by TRIM (Transport of Ions in Matter) simulations indicate that with increasing the fluence of 10-MeV protons from $1 \times 10^{12}$ to $1 \times 10^{15}$ $n_{eq}/cm^2$, the concentration of divacancies ($V_2$) in silicon rises from ~$1 \times 10^{12}$ to ~$1 \times 10^{15}$ cm$^{-3}$ [1]. The concentrations of trivacancies ($V_3$) after irradiation with 10-MeV protons under fluences of $1 \times 10^{13}$ and $1 \times 10^{15}$ $n_{eq}/cm^2$ are ~$5 \times 10^{12}$ and ~$3 \times 10^{14}$ cm$^{-3}$, respectively [1]. The most likely process of small vacancy cluster formation is by adding to another vacancy, or to previously formed $V_2$, $V_3$, and $V_4$ aggregates, or by a single vacancy migrating after kicking off a Si atom from a substitutional position [1,4,5]. The single vacancy diffuses much faster than the vacancy aggregates composed of two, three, or four vacancies, and can be easily captured by a larger defect [3–5]. On the other hand, the vacancy aggregates can dissociate into smaller parts and the energies needed for the dissociation processes $V_2 \rightarrow V + V$, $V_3 \rightarrow V_2 + V$, $V_4 \rightarrow V_3 + V$, and $V_5 \rightarrow V_4 + V$, predicted by molecular-dynamics (MD) calculations, are 1.69, 2.08, 1.92, and 3.05 eV, respectively [5].

The energy levels of the bandgaps of small vacancy clusters in irradiated silicon are known only for divacancies and trivacancies. These levels are involved in the charge carrier capture and emission processes. The electronic properties of divacancies have been extensively studied since 1961, when Corbett and Watkins experimentally identified the divacancy center in electron-irradiated silicon from the electron spin resonance (ESR) spectra Si-*G6* and Si-*G7* [4,6]. The studies performed by deep-level transient spectroscopy (DLTS) have shown that in silicon irradiated with hadrons, $V_2$ may be observed in four different charge states: (+), (0), (−), and (2−) [3,4]. In other words, $V_2$ can interact with excess charge carriers in three ways. Firstly, it can be a singly ionized donor center acting as a hole trap with the energy level at ~0.21 eV above the valence band edge ($E_v$) and the thermal emission of holes from this level leads to a $V_2$ (+/0) charge state change [2,4]. Secondly, $V_2$ can form a singly ionized acceptor center being an electron trap characterized by the energy level at ~0.43 eV below the conduction band edge ($E_c$) and the $V_2$ (−/0) charge state change occurs due to the thermal emission of electrons from this level [4,7]. Finally, $V_2$ can also form a doubly ionized acceptor center being an electron trap characterized by the energy level at ~0.24 eV below $E_c$ and the thermal emission of electrons from this level is related to the $V_2$ (2−/−) charge state change [4,7].

The electronic properties of trivacancies in Si have been much less studied compared to those of divacancies. According to the ESR results obtained by Lee and Corbett, who attributed the Si-A4 ESR spectrum to $V_3$ in neutron-irradiated silicon, the $V_3$ aggregate structure is consistent with a (110) planar configuration of vacancies forming part of a hexagonal ring (PHR) [8]. The DLTS experiments, performed many decades later [9], showed that $V_3$ in this configuration gives rise to two acceptor levels at ~360 and 460 meV below $E_c$, corresponding to the charge state changes of $V_3$ (2−/−) and $V_3$ (−/0), respectively. It is worth noting that the difference between the activation energies for electron emission from $V_3$ (2−/−) and $V_2$ (2−/−) is fairly large, but the difference between the activation energies for electron emission from $V_3$ (−/0) and $V_2$ (−/0) is very small, being in the experimental error margin of the conventional DLTS technique [9]. This fact is consistent with the results of calculations [5,10] predicting small differences between the energy level positions related to the (−/0) charge state change for the clusters with the number of vacancies rising from 2 to 5. The resolution of the conventional DLTS technique was sufficiently high to determine the $V_2$ (−/0) energy level, but the separation of the electron thermal emission signals for $V_2$ (−/0) and $V_3$ (−/0), as well as for $V_2$ (−/0) and $V_3$ (2−/−), was possible by using Laplace DLTS, which was able to determine the number of exponential components with different time constants in the capacitance relaxation waveform measured at a given temperature [9]. In this way, the $E_c - 360$ and $E_c - 460$ meV levels corresponding to $V_3$ (2−/−) and $V_3$ (−/0), respectively, were found [9].

The aim of this study is to determine the characteristics and concentrations of deep traps in in high-purity silicon (HPSi) irradiated with high fluences of 23-MeV protons, and to identify these traps with the energy levels of $V_2$, $V_3$, $V_4$, and $V_5$ clusters using

the available results on the properties of radiation defects in silicon obtained by DLTS or theoretical calculations. The deep trap characteristics in the as-irradiated semi-insulating material are studied by state-of-the-art Laplace photoinduced transient spectroscopy, also known as high-resolution photoinduced transient spectroscopy (HRPITS), which is capable of detecting of charge carrier traps with slightly different activation energies and capture cross-sections in semi-insulating silicon produced by irradiation with hadron fluences above $1 \times 10^{13}$ $n_{eq}/cm^2$ [11–13]. We have been strongly motivated by the results of theoretical calculations showing that, similar to $V_2$ and $V_3$ clusters, higher order clusters, such as $V_4$ and $V_5$, can introduce into the Si bandgap the acceptor levels related to the charge state changes of $V_4$ $(2-/-)$, $V_4$ $(-/0)$, $V_5$ $(2-/-)$, and $V_5$ $(-/0)$ involving thermal emission of electrons [5,10].

## 2. Materials and Methods

For characterization of irradiation-induced deep traps we have used HPSi wafers of 100 mm in diameter fabricated from an *n*-type single crystal grown in the <100> direction by the floating zone (FZ) method. The resistivity of the ingot exceeded 5000 $\Omega$cm, and in order to get the uniformly distributed target resistivity of ~2000 $\Omega$cm, the ingot was subjected to doping with phosphorus (P) in the neutron transmutation doping (NTD) process. The P concentration in the ingot after the NTD process was ~$2.4 \times 10^{12}$ $cm^{-3}$. The residual oxygen concentration ([O]) in the HPSi crystal was ~$5 \times 10^{15}$ $cm^{-3}$. This value was derived from the absorption coefficient measurements at the wavenumber of 1136 $cm^{-1}$ performed by the Fourier transform infrared (FTIR) spectroscopy at 4 K. The residual carbon concentration ([C]) was found to be below the detection limit of ~$5 \times 10^{15}$ $cm^{-3}$ for the FTIR measurements of the absorption line at the wavenumber of 608 $cm^{-1}$ at 4 K. The residual nitrogen concentration ([N]) was also found to be below the detection limit, equal to $2 \times 10^{14}$ $cm^{-3}$, from the FTIR measurements of the height of the absorption peak occurring at the wavenumber of 963 $cm^{-1}$ at 300 K [12]. The residual boron concentration determined by low-temperature FTIR spectroscopy was ~$5 \times 10^{11}$ $cm^{-3}$. The minority charge carrier lifetime was ~2 ms which indicates that the concentration of metal impurities serving as recombination centers was below ~$5 \times 10^{13}$ $cm^{-3}$.

The thickness of the device-quality HPSi wafers used in the experiment was 300 μm. The upper (100) surface was mirror-like polished, and the bottom one was ground and etched. Before proton irradiation, the wafers were cut into chips with dimensions of $1 \times 1$ $cm^2$. The (100) surface of the HPSi chips was irradiated with 23-MeV protons in a state-of-the-art accelerator at the Karlsruhe Institute of Technology (KIT). The chips were irradiated with four different proton fluences equivalent to irradiation with 1-MeV neutrons in terms of non-ionizing energy loss (NIEL) displacement damage, namely: $1 \times 10^{14}, 5 \times 10^{14}, 1 \times 10^{15}$, and $5 \times 10^{15}$ $n_{eq}/cm^2$. During the irradiation the proton beam current was in the range of 1.5–1.7 μA and the chip temperature did not exceed room temperature. After the irradiation, the chips were always stored at $-20$ °C, except for a time, estimated as ~100 h, when they were kept at room temperature during transportation, preparation of ohmic contacts, and measurements. According to the experimental studies reported by Watkins and molecular dynamics simulations, the small vacancy aggregates are considered to be stable defects at room temperature [1,5].

For the Laplace PITS measurements, arrays of two co-planar Al ohmic contacts were evaporated through a metal mask. The contacts were in the shape of squares of $2.5 \times 2.5$ $mm^2$ in area. The gap between them for the material illumination to generate excess charge carriers was 0.7 mm. Single samples with dimensions of $4 \times 9$ $mm^2$ were cut from the chips and mounted on the cold finger of an optical cryostat. The experimental set-up included a Leybold closed-cycle helium optical cryostat (Cologne, Germany), a Power Technology metrological semiconductor laser (Alexander, AR, USA) with a beam spectral line at 690 nm (photon energy 1.8 eV) and 30-mW beam power, a Keithley 428 fast current amplifier (conductance voltage converter) (Keithley Instruments, Cleveland, OH, USA), and a digital acquisition unit, allowing the digitization of the photocurrent transients, with a 12-bit amplitude resolution and a 1-μs time resolution. Before recording the

photocurrent transients, the irradiated material resistivity and the activation energy of the dark conductivity ($E_{TDC}$) were determined from the measurements of the temperature dependence of dark current. Additionally, the mobility-lifetime ($\mu\tau$) product as a function of temperature was determined by using the constant photocurrent method (CPM) in the pulse mode [12,13]. This method is based on the assumption that the $\mu\tau$ product at a given temperature is proportional to the height of the photocurrent pulse at the end of the optical excitation pulse. For $\alpha d \gg 1$, where $\alpha$ is the absorption coefficient and $d$ is the sample thickness, the $\mu\tau$ product can be defined by the relation [12,13]:

$$\mu\tau = I_{\text{ph}}/[\eta(1-R)qlEF],\tag{1}$$

in which $I_{\text{ph}}$ is the height of the photocurrent pulse generated by the photon flux $F$, $\eta$ is the quantum efficiency of the electron-hole pairs generation, $R$ is the reflection coefficient, $q$ is the elementary charge, $l$ is the contacts width, and $E$ is the applied electric field dependent on the voltage applied between the two contacts on a sample. The photon flux, controlled by dedicated optical filters, was $4.6 \times 10^{14}$ cm$^{-2}$s$^{-1}$ and the electric field was 286 V/cm. The values of $\eta$ and $R$ were assumed to be 1 and 30%, respectively. The excitation pulse width and the repetition period were 50 ms and 500 ms, respectively.

The Laplace PITS measurements are based on recording the photocurrent transients, generated by optical pulses, and an advanced numerical analysis of the photocurrent relaxation waveforms observed after switching off the light. To improve the signal to noise ratio, digital data were averaged, usually taking 500 transients. To determine the parameters of radiation defect centers, the photocurrent transients were measured at the photon flux values selected accordingly to the proton fluence used for the irradiation. Namely, for the samples irradiated with fluences of $1 \times 10^{14}$ and $5 \times 10^{14}$ $n_{\text{eq}}$/cm$^2$, a photon flux of $1.4 \times 10^{15}$ cm$^{-2}$s$^{-1}$ was used, and for the samples irradiated under fluences of $1 \times 10^{15}$ and $5 \times 10^{15}$ $n_{\text{eq}}$/cm$^2$, the photon flux was higher, equal to $4.5 \times 10^{17}$ cm$^{-2}$s$^{-1}$. Using the larger photon flux for illuminating the samples irradiated with the higher fluences enabled the similar ratio of the relaxation waveform amplitude to the photocurrent pulse amplitude to be kept for the samples irradiated with the lower and higher fluences. The voltage applied between the two coplanar contacts was 20 V. For further processing, each photocurrent transient was normalized with respect to the photocurrent pulse height at the end of the illumination pulse. The transients were digitally recorded in a broad temperature range of 100–320 K in steps of 2 K. In order to get the temperature dependences of the thermal emission rate of charge carriers for detected defect levels, a two-dimensional (2D) analysis of the photocurrent decays as a function of time ($t$) and temperature ($T$) was carried out.

In the Laplace PITS, the analysis of the of photocurrent relaxation waveforms is performed under the assumption that each of these waveforms observed at a given temperature $T$ is the sum of a number of exponential components produced by the thermal emission of the excess charge carriers from various energy levels of defect centers [11,12]. The time constants of these exponential components are equal to the reciprocal values of the charge carrier thermal emission rate ($e_{\text{T}}$) for the defect centers. The temperature dependences of the emission rate for the defect centers are extracted from the photocurrent relaxation waveforms by using a two-dimensional (2D) numerical procedure based on the inverse Laplace transform algorithm implemented in the CONTIN code developed by S.W. Provencher [12]. First, from each photocurrent relaxation waveform recorded at a given temperature $T_j$, the one-dimensional (1D) Laplace spectrum with sharp peaks at various values of the thermal emission rate characteristic for the defect centers detected at this temperature is obtained. Next, the 2D Laplace spectrum, with the spectral fringes for the defect centers indicating the temperature changes of the emission rate, is created by putting together all the 1D Laplace spectra resulting from the analysis of the photocurrent relaxation waveforms recorded at all temperatures ($j = 1, 2, 3 \ldots n$). Finally, the 2D Laplace spectral fringes for all the defect centers detected are visualized in 3D space as the sharp folds. For convenience, these folds are projected on the plane determined by the axes ($T$, $e_{\text{T}}$) and the projected ridgelines of the folds depict the temperature dependences of the charge

carrier emission rate $e_T(T)$ [11]. These data are used to make the Arrhenius plots, being the signatures of defect centers. The experimental data forming these plots are fitted with a straight line by means of the linear regression, and the defect center's parameters are determined according to the Arrhenius equation:

$$e_T(T) = AT^2 \exp(-E_a/k_B T), \tag{2}$$

where $E_a$ is the activation energy, $k_B$ is the Boltzmann constant, $A = \gamma \sigma_a$ is the pre-exponential factor equal to the product of the material constant $\gamma$, dependent on the effective mass and the apparent capture cross-section for electrons or holes, $\sigma_a$. Thus, each defect center is characterized by the activation energy $E_a$ of the electron or hole thermal emission and pre-exponential factor $A$. According to our calculations, the values of $\gamma_n$ and $\gamma_p$, necessary to get the apparent capture cross-sections for electrons or holes, are $7.17 \times 10^{21}$ and $3.80 \times 10^{21}$ $K^{-2} cm^{-2} s^{-1}$, respectively.

The procedure developed for determining the concentration ($N_T$) of a defect center is based on a model that the amplitude of the photocurrent relaxation waveform for a single defect center determined at a given temperature $T$ is proportional to the concentration, $n_T(0)$, of electrons or holes trapped by this center when switching off the optical excitation pulse [12,13]. This procedure is executed in three steps. First, the amplitude $I(0)$ of the exponential signal produced by the thermal emission of charge carriers at the temperature $T$ for a given type of defect centers is determined. This amplitude has been experimentally found to be related to the height of the Laplace peak, $I_L$, according to the formula: $I(0) = 2.7 \, I_L \times I_{ph}$, where $I_{ph}$ is the height of the photocurrent pulse at the moment of optical excitation pulse termination [12]. This empirical formula is valid for the emission rate values below $1300 \ s^{-1}$. The coefficient has been established from the plot of the $I(0)$ experimental values against the $I_L \times I_{ph}$ product. The linear regression of this dependence gave the slope of $2.7 \pm 0.13$ with $R^2 = 0.96$. In the second step, the $n_T(0)$ at temperature $T$ is calculated using the formula

$$n_T(0) = I(0)/(q e_T E C \mu \tau), \tag{3}$$

where $C$ is a geometrical parameter representing the area of the cross-section of a sample region through which the excess charge carriers emitted from a defect center flow to the electrodes on the sample surface. This parameter is dependent on temperature according to the formula:

$$C(T) = (1/\alpha + L_D)l, \tag{4}$$

where $\alpha$ is the dependent on temperature absorption coefficient, $l$ is the electrodes width, and $L_D$ is the temperature dependent charge carrier diffusion length, given by $L_D = [(k_B T/q)\mu\tau]^{1/2}$. In the final step, the $n_T(0)$ and $e_T$ values are calculated for several $T$ values using Equations (2)–(4), and the dependence of $n_T(0)$ as a function of $e_T$ is plotted. The trap concentration, $N_T$, is determined by extrapolation of the $n_T(0) = f(e_T)$ plot to the value of $n_T(0)$ for $e_T \to 0$, according to the formula [12,13]:

$$n_T(0) = N_T/(1 + e_T/G\tau c_T), \tag{5}$$

where $G$, $\tau$, and $c_T$ are the excess charge carrier generation rate, lifetime, and capture coefficient, respectively.

## 3. Results

### 3.1. Energy Levels

By recording the photocurrent transients at temperatures in the range of 100–320 K, we received a set of data allowing determination of the characteristics of charge carriers traps using the CONTIN procedure. For the HPSi samples irradiated with each fluence, these being $1 \times 10^{14}$, $5 \times 10^{14}$, $1 \times 10^{15}$, and $5 \times 10^{15}$ $n_{eq}/cm^2$, the peaks in the Laplace spectra, derived from the analysis of photocurrent relaxation waveforms recorded at the

same temperatures, were observed at the same values of the thermal emission rate of charge carriers. In other words, for each fluence, the same traps related to the radiation defect centers were observed. Figures 1 and 2 illustrate the one-dimensional Laplace spectra obtained by using the CONTIN numerical procedure to extract the exponential components in the photocurrent relaxation waveforms recorded at temperatures of 211.95 and 255.15 K, respectively, for a sample irradiated with a fluence of $1 \times 10^{14}$ $n_{eq}/cm^2$.

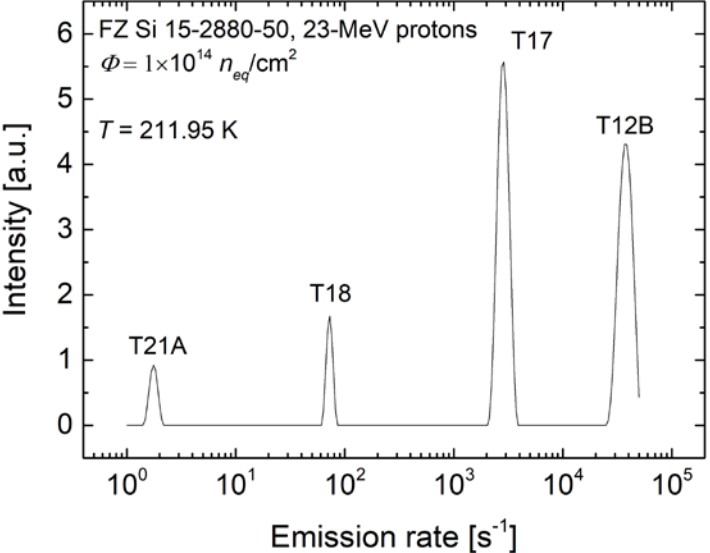

**Figure 1.** Laplace spectrum derived from the photocurrent relaxation waveform recorded at 211.95 K for a HPSi sample after irradiation with a fluence of $1 \times 10^{14}$ $n_{eq}/cm^2$ of 23-MeV protons. Four sharp peaks indicating the thermal emission rate values for the irradiation-induced deep traps labeled as T12B, T17, T18, and T21A are visible.

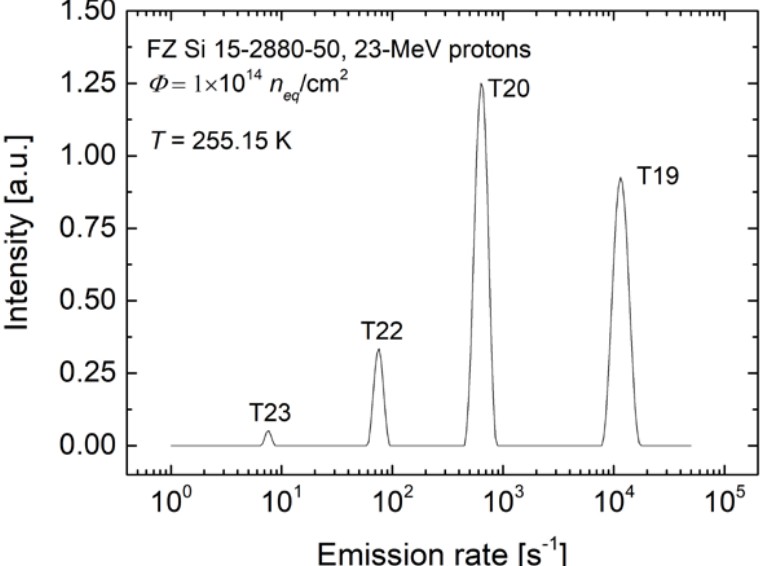

**Figure 2.** Laplace spectrum derived from the photocurrent relaxation waveform recorded at 255.15 K for a HPSi sample irradiated with a fluence of $1 \times 10^{14}$ $n_{eq}/cm^2$ of 23-MeV protons. Four sharp peaks indicating the thermal emission rate values for the irradiation-induced deep traps labeled as T19, T20, T22, and T23 are visible.

The one-dimensional Laplace spectrum in Figure 1 consists of four sharp peaks indicating the thermal emission of charge carriers from the four irradiation-induced deep traps, labelled as T12B, T17, T18, and T21A. The charge carrier emission rates of the traps

at 211.95 K, corresponding to the peaks positions on the x-axis, are 36,900, 2750, 72.6, and 1.83 s$^{-1}$, respectively. According to Equation (2), among these traps the highest value for the activation energy for thermal emission of charge carriers at 211.95 K is expected for the T21A trap, and the lowest value for the activation energy will be for the T12B trap. In the one-dimensional Laplace spectrum obtained from the analysis of the photocurrent relaxation waveform recorded for the same HPSi sample at 255.15 K, the four peaks labelled as T19, T20, T22, and T23 corresponding to the other four irradiation-induced deep traps is seen. At 255.15 K, the charge carrier emission rates for the T19, T20, T22, and T23 traps are 11,500, 632, 75.8, and 7.65 s$^{-1}$, respectively. Among these irradiation-induced defect centers, the T19 trap is the shallowest and the T23 trap is the deepest.

The T12B trap is likely to be attributed to the divacancy-related level involving the thermal electron emission from a doubly negative divacancy. The energy level corresponding to the charge state change of $V_2$ (2−/−), revealed from DLTS measurements, is located in the Si bandgap at $E_c − (0.24 ± 0.01)$ eV [3,7,14,15]. The reported DLTS spectra, however, are presented in a very small emission rate window of 2.56 s$^{-1}$, so the position of the $V_2$ (2−/−) peak on the temperature axis is observed at a much lower temperature, equal to ~114 K [7]. The T17 trap, whose thermal emission rate of charge carriers at 211.95 K is 2750 s$^{-1}$ (Figure 1), can be identified with the trivacancy-related level at $E_c − 0.359$ eV assigned to the electron emission leading to the charge state change $V_3$ (2−/−) [9]. The measurements of the reported Laplace DLTS spectra indicating the thermal electron emission from this energy level were performed at an emission rate window of 80 s$^{-1}$ [9], so the position of the $V_3$ (2−/−) peak on the temperature axis was at ~185 K. The T18 and T21A traps, located deeper in the bandgap compared to the T17 trap, can be attributed to the charge state changes of $V_4$ (2−/−) and $V_5$ (2−/−), respectively [5,10,15]. The results of theoretical analysis indicate that the higher order clusters, such as $V_4$ and $V_5$, can introduce the acceptor levels related to the charge state changes of $V_4$ (2−/−), $V_4$ (−/0), $V_5$ (2−/−), and $V_5$ (−/0) involving thermal emission of electrons into the Si bandgap [5,10,15]. These results also show that the small vacancy clusters are not the negative $U$ centers, and the following relations between the activation energies $E_a$ for electron thermal emission from these centers are expected: $E_a$ [$V_2$ (2−/−)] < $E_a$ [$V_2$ (−/0)], $E_a$ [$V_3$ (2−/−)] < $E_a$ [$V_3$ (−/0)], $E_a$ [$V_4$ (2−/−)] < $E_a$ [$V_4$ (−/0)], and $E_a$ [$V_5$ (2−/−)] < $E_a$ [$V_5$ (−/0)]. Moreover, two additional relations should be fulfilled: $E_a$ [$V_2$ (2−/−)] < $E_a$ [$V_3$ (2−/−)] < $E_a$ [$V_4$ (2−/−)] < $E_a$ [$V_5$ (2−/−)] and $E_a$ [$V_2$ (−/0)] < $E_a$ [$V_3$ (−/0)] < $E_a$ [$V_4$ (−/0)] < $E_a$ [$V_5$ (−/0)] [9]. In other words, for the higher vacancy cluster order, the energy level positions associated with the (2−/−) or (−/0) charge states changes are deeper in the Si bandgap.

At 255.15 K, the charge carrier emission rates for the T19, T20, T22, and T23 traps (Figure 2) are 11,500, 632, 75.8, and 7.65 s$^{-1}$, respectively. Among these irradiation-induced defect centers, the T19 trap is the shallowest one and the T23 trap has the highest activation energy for charge carrier emission. The T19 trap can be tentatively assigned to the energy level corresponding to the charge state change of $V_2$ (−/0) involving the thermal emission of electrons from singly ionized divacancies. According to the reported Laplace DLTS spectrum for a silicon sample irradiated with high-energy electrons [9], the electron emission rate at 230 K for this charge state change of divacancies is ~200 s$^{-1}$ and at 255.15, it can reach 11,500 s$^{-1}$. Similarly, the T20 trap can be tentatively assigned to the energy level related to the charge state change of $V_3$ (−/0) involving the thermal emission of electrons from singly ionized trivacancies. The Laplace DLTS results indicate that the electron emission rate at 230 K for this charge state change of trivacancies is ~80 s$^{-1}$ and at 255.15, it can reach 632 s$^{-1}$ [9]. On the grounds of the theoretical predictions mentioned above [5,10,15], the traps T22 and T23 with low values of emission rate at 255.15 K can be tentatively assigned to the energy levels located near the middle of the bandgap related to the charge state changes of $V_4$ (−/0) and $V_5$ (−/0), respectively. It should be noted that $V_4$ aggregates have been predicted theoretically [5,15] to be in the most energetically favored configuration when the four Si atoms are removed from the hexagonal ring in the silicon lattice. Experimentally, they were assigned to the Si-*P3* center observed by ESR in silicon irradiated with

a high fluence of fast neutrons [8]. The formation of $V_4$ aggregates has been also found by positron annihilation spectroscopy (PAS) in high-purity *n*-type silicon irradiated with high-energy oxygen ions at room temperature up to a fluence of $5 \times 10^{15}$ ions/cm$^2$ [16]. The results of PAS studies have shown that the positron lifetime value of $338 \pm 10$ ps is in very good agreement with the values obtained by theoretical calculations [16]. So far, however, the energy levels for the tetravacancies have not been unambiguously determined experimentally. This lack of results is because $V_4$ aggregates arise in silicon irradiated with high fluences of high-energy particles, and the DLTS technique is not applicable to studies of defect levels in the semi-insulating material with a resistivity of ~$10^5$ $\Omega$cm formed under such irradiation conditions [11,13]. This method is based on the capacitance measurements and their accuracy, as well sensitivity, which is strongly affected by the samples series resistance [2]. The $V_5$ aggregates, similar to tetravacancies and trivacancies, may form stable configurations in the silicon lattice that have been identified using density functional theory (DFT) calculations [5,10]. It has been demonstrated that these configurations are in the form of complex defects made up of a ring including six vacancies and one, two, or three self-interstitials, respectively [5,10]. These complexes have no dangling bonds and they are expected to be more stable than the PHR configurations, where two dangling bonds remain at the ends of the vacancy chain [17]. Experimentally, the $V_5$ aggregate has been identified from the Si-*P*1 spectrum recorded by ESR for silicon irradiated with a high fluence (~$10^{18}$ $n$/cm$^2$) of fast neutrons [18]. The structural model based on these results assumed that three vacancies are aligned in a row along the <110> direction, with two additional vacancies bound to the vacancies located at the ends of both sides of the vacancy chain [18]. The PAS results indicate [19] that the positron lifetime associated with the $V_5$ cluster is 390 ps and this value is consistent with that of 376 ps; obtained for this defect by theoretical calculations performed by Puska and Corbel [20]. The $V_5$ aggregate energy levels, similar to those of the $V_4$ cluster, have not been determined yet due to the high resistivity of silicon irradiated with high fluences of hadrons [11,18].

The Arrhenius plots illustrating the dependence of the emission rate of charge carriers as a function of the thermal energy for all the traps detected by the Laplace PITS measurements using the irradiated HPSi samples are presented in Figure 3. These plots, the slopes of which reflect the activation energy values and the intercepts of which allow determination of the values of the capture cross-section, are unique signatures of defect centers and can be used for their identification. Therefore, in Figure 3a, the Arrhenius plots for T12B and T17 traps are compared with those established earlier by DLTS measurements [7,9] for the $V_2$ ($2-/-$) and $V_3$ ($2-/-$) levels, respectively. The reference Arrhenius plots are marked with the broken lines. It is seen that the slopes of the plots for the T12B trap and $V_2$ ($2-/-$) level are slightly different, and this difference is reflected in the activation energy values. The activation energy value for the former is 0.255 eV and, according to the data in [7], the activation energy value for the latter is 0.24 eV. The difference between the values is 0.015 eV, which represents ~6.3% of the value for $V_2$ ($2-/-$). Thus, the T12B trap can be assigned to the $V_2$ ($2-/-$) level. The Arrhenius plots for the T17 trap and $V_3$ ($2-/-$) level perfectly match, and provide strong evidence for attributing this trap to the charge state change of the doubly ionized trivacancy induced by electron thermal emission. Taking into account the results of theoretical analysis as well as experimental studies [2,10,15,21], the Arrhenius plots for the T18 and T21A traps are postulated to be assigned to the $V_4$ ($2-/-$) and $V_5$ ($2-/-$) levels, respectively. In Figure 3b, the Arrhenius plots for the T19 and T20 traps are compared with those which have been previously established by DLTS measurements [7,9], marked with broken lines, for the $V_2$ ($-/0$) and $V_3$ ($-/0$) levels, respectively. The presented results indicate that there is an excellent match in the signatures for the T19 trap and the $V_2$ ($-/0$) level, as well as in the signatures of the T20 trap and the $V_3$ ($-/0$) level. This fact allows us to attribute the T19 and T20 traps to the energy levels of the neutralizations of a singly ionized divacancy and a singly ionized trivacancy, respectively, induced by the thermal emission of electrons. The Arrhenius plots the T22 and T23 traps are postulated to be assigned to the $V_4$ ($-/0$) and $V_5$ ($-/0$) levels, respectively.

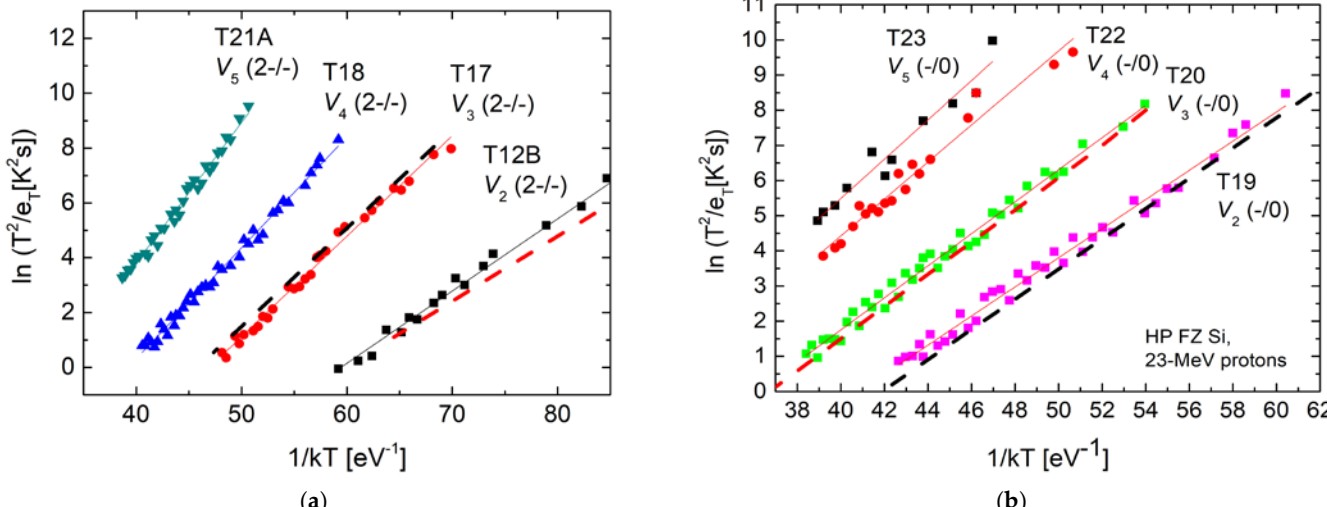

**Figure 3.** Arrhenius plots showing the temperature dependence of the thermal emission rate of charge carriers for the traps formed in HPSi under 23-MeV proton irradiation with fluences of $1 \times 10^{14}$, $5 \times 10^{14}$, $1 \times 10^{15}$, and $5 \times 10^{15}$ $n_{eq}/cm^2$. (**a**) The Arrhenius plots for the T12B (black squares), T17 (red circles), T18 (blue triangles), and T21A (dark green inverted triangle) traps. The broken lines in red and black mark the Arrhenius plots reported for the $V_2$ $(2-/-)$ and $V_3$ $(2-/-)$ levels [7,9] for comparison with those for the T12B and T17 traps, respectively. (**b**) The Arrhenius plots for the T19 (magenta squares), T20 (light green squares), T22 (red circles), and T23 (black squares) traps. The broken lines in black and red mark the Arrhenius plots reported for the $V_2$ $(-/0)$ and $V_3$ $(-/0)$ levels [7,9] for comparison with those for the T19 and T20 traps, respectively.

The values of the thermal activation energy and pre-exponential factor from the Arrhenius equation, established for all the traps detected in the proton irradiated HPSi samples from the slope and intercept of each line shown in Figure 3, are listed in Table 1. The traps identification with the changes in the charge states of small vacancy clusters in silicon is also shown in this table and the values of electron capture cross-section corresponding to this identification are also included. The arrangement of the deep traps T12B, T17, T18, T19, T20, T21A, T22, and T23 in the Si bandgap tentatively attributed to the energy levels related to the small vacancy cluster charge state changes is shown in Figure 4. The results indicate that the energy positions of the T12B, T17, T18, and T21A traps follow the trend theoretically predicted for the energy levels of $V_2$ $(2-/-)$, $V_3$ $(2-/-)$, $V_4$ $(2-/-)$, and $V_5$ $(2-/-)$ [10]. Simultaneously, the T19, T20, T22, and T23 traps are in line with the theoretical predictions for the energy levels of $V_2$ $(-/0)$, $V_3$ $(-/0)$, $V_4$ $(-/0)$, and $V_5$ $-/0)$ [9,10,15].

**Table 1.** Values of the thermal emission activation energy ($E_a$), pre-exponential factor in the Arrhenius equation ($A$), and apparent electron capture cross-section for deep traps found by Laplace PITS in HPSi irradiated with 24-MeV protons. The assignment of traps to the energy levels of small vacancy clusters formed under the irradiation is summarized.

| Trap Label | $E_a$ [meV] | $A$ [$K^{-2}s^{-1}$] | $\sigma_{na}$ [1] [$cm^2$] | Trap Identification |
|---|---|---|---|---|
| T12B | $255 \pm 10$ | $9.7 \times 10^6$ | $1.35 \times 10^{-15}$ | $V_2$ $(2-/-)$ |
| T17 | $367 \pm 5$ | $3.0 \times 10^7$ | $4.18 \times 10^{-15}$ | $V_3$ $(2-/-)$ |
| T18 | $405 \pm 5$ | $7.8 \times 10^7$ | $1.09 \times 10^{-14}$ | $V_4$ $(2-/-)$ |
| T19 | $415 \pm 5$ | $2.2 \times 10^7$ | $3.07 \times 10^{-15}$ | $V_2$ $(-/0)$ |
| T20 | $456 \pm 5$ | $1.5 \times 10^7$ | $2.09 \times 10^{-15}$ | $V_3$ $(-/0)$ |
| T21A | $512 \pm 10$ | $1.8 \times 10^7$ | $2.51 \times 10^{-15}$ | $V_5$ $(2-/-)$ |
| T22 | $526 \pm 10$ | $1.7 \times 10^7$ | $2.37 \times 10^{-15}$ | $V_4$ $(-/0)$ |
| T23 | $559 \pm 10$ | $2.1 \times 10^7$ | $2.93 \times 10^{-15}$ | $V_5$ $(-/0)$ |

[1] It is assumed that the material constant $\gamma_n$ in the product $A = \gamma_n \sigma_n$ is $7.17 \times 10^{21}$ $K^{-2}s^{-1}cm^{-2}$.

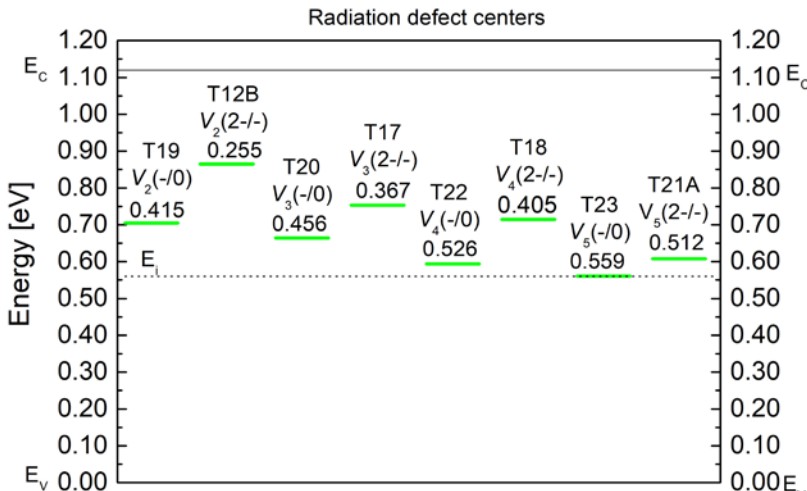

**Figure 4.** Positions of the T12B, T17, T18, and T21A traps attributed to the energy levels in the Si bandgap related to the (2−/−) charge state change of divacancies, trivacancies, tetravacancies, and pentavacancies together with the positions of the T19, T20, T22, and T23 traps attributed to the energy levels corresponding to the (−/0) charge state change of these defects. The numbers give the energy in electronvolts with respect to the conduction band minimum.

The reported values for the activation energy for electron emission associated with the transitions of $V_2$ (2−/−) and $V_2$ (−/0) are (0.23–0.24) eV and (0.42–0.43) eV, respectively [2,3,7,14]. These data have been obtained by DLTS measurements with an error of around 10%. The activation energies of 255 and 415 meV shown in Figure 4 for the T12B and T19 traps are extremely consistent with those known for the divacancies. It worth adding that the activation energy value for the T19 trap is smaller than that of 0.43 eV reported in [7] for $V_2$ (−/0). The difference is 0.015 eV, which represents ~3.5% of the activation energy value for $V_2$ (−/0). The activation energies of the T17 and T20 traps perfectly match those determined by Laplace DLTS for $V_3$ (2−/−) and $V_3$ (−/0), equal to 359 and 458 meV, respectively [9].

Radiation defects with higher activation energies for electron emission, ranging from 0.5 to 0.61 eV, have been observed in silicon by conventional DLTS [21,22]. The measurements revealed a midgap trap at $E_c − 0.61$ eV, formed in slightly phosphorus-doped FZ Si under irradiation with 61-MeV protons [22] as well as two midgap traps, at $E_c − 0.56$ eV and $E_c − 0.61$ eV, formed in *n*-type Czochralski-grown Si due to the implantation of $^{28}Si^{3+}$ ions with energy of 8.3 MeV at room temperature [21]. The energy levels for $V_2$ (2−/−), $V_2$ (−/0), $V_3$ (2−/−), $V_3$ (−/0), $V_4$ (2−/−), $V_4$ (−/0), $V_5$ (2−/−), and $V_5$ (−/0) have been theoretically estimated using the defect-molecule model assuming that the vacancy aggregates are in zig-zag configuration and the two Si atoms at the ends of the chain of vacancies in (110) plane form a molecular bond in the <111> direction [10,17]. These levels were found to be dependent on the distance between the Si atoms at both ends of the multi-vacancy chain. This distance was calibrated by the number of vacancies between these Si atoms [10]. For the calculations, both the charge state changes and the correlation effects were taken into account. The results show that the small vacancy clusters can be represented in terms of a family of defects whose electronic properties consecutively change with increasing the number of vacancies in the multi-vacancy chain. Thus, the small vacancy clusters are not the negative *U* centers, and the energy levels of $V_2$ (2−/−), $V_3$ (2−/−), $V_4$ (2−/−), and $V_5$ (2−/−) are shallower than that of $V_2$ (−/0), $V_3$ (−/0), $V_4$ (−/0), and $V_5$ (−/0), respectively. In addition, the higher the order of the vacancy cluster, the deeper the cluster energy level position in the Si bandgap [5,10]. It is easily seen that the experimental results visualized in Figure 4 are fully consistent with the arrangement of the small vacancy cluster energy levels in the Si bandgap predicted theoretically. The formation energies of $V_3$, $V_4$, and $V_5$ defects, as well as the $V_4$ (2−/−), $V_4$ (−/0), $V_5$ (2−/−), and $V_5$ (−/0) energy levels,

have also been calculated by using density functional theory (DFT) together with the local spin density approximation (LSDA) for the exchange and correlation potential [15]. The results of these calculations indicate that the formation energy values for the $V_3$, $V_4$, and $V_5$ clusters in the Four-Fold-Coordinated (FFC) configurations are lower than those in the PHR structures by 0.6 eV, 1.1 eV, and 0.7 eV, respectively [15]. The positions of deep acceptor levels for $V_4$ (2−/−) and $V_4$ (−/0) in the FFC configuration have been predicted to be the same, at $E_c − 0.54$ eV [15]. The energy levels of $V_5$ (2−/−) and $V_5$ (−/0) in the same configuration are found to be located at $E_c − 0.47$ eV and $E_c − 0.45$ eV, respectively [15]. These results show that the energy level at $E_c − 0.54$ eV, derived theoretically for $V_4$ (−/0), is very close to that at $E_c − 526$ meV, determined experimentally for the T22 trap (Table 1, Figure 4). On the other hand, the calculated level at $E_c − 0.54$ eV for $V_4$ (2−/−) seems to be too deep in view of the results of other theoretical studies [10], as well as compared to our experimental result for the T18 trap (Table 1, Figure 4). As far as the energy levels for the pentavacancies are concerned, the close values between the calculated level at $E_c − 0.47$ eV for $V_5$ (2−/−), and that at $E_c − 512$ meV determined experimentally for the T21A trap are observed. However, in view of other theoretical works [5,10,17], the energy level at $E_c − 0.45$ eV calculated for $V_5$ (−/0) [15] should be located closer to the midgap. The energy level at $E_c − 559$ meV determined experimentally for the T23 trap corresponds well with these suggestions.

### 3.2. Defect Concentrations

The concentrations of the T12 B and T19 traps, assigned to $V_2$ (2−/−) and $V_2$ (−/0), respectively, as well as those of the T17 and T20 traps, assigned to $V_3$ (2−/−) and $V_3$ (−/0), respectively, are plotted against the fluence ranging from $1 \times 10^{14}$ to $5 \times 10^{15}$ $n_{eq}/cm^2$ in Figure 5. Similarly, the concentrations of the T18 and T22 traps, attributed to the charge state changes of tetravacancies $V_4$ (2−/−) and $V_4$ (−/0), respectively, as well as those of the T21A and T23 traps, assigned to the charge state changes of pentavacancies $V_5$ (2−/−) and $V_5$ (−/0), respectively, are shown as a function of the fluence in Figure 6. According to the plots presented in these figures, the concentrations of all kinds of vacancy aggregates increase with a rise in the fluence. This fact indicates that they are formed directly under proton irradiation of HPSi samples at room temperature, and the increase in the fluence by an order of magnitude, from $1 \times 10^{14}$ to $1 \times 10^{15}$ $n_{eq}/cm^2$, is sufficient to produce significant concentrations of small vacancy clusters. In other words, during irradiation two processes take place: formation of vacancies through the displacement of silicon atoms from substitutional positions and interaction between vacancies leading to the formation of various types of their aggregates.

To understand the results shown in Figures 5 and 6, a comment explaining how the defect's charge state changes during the HRPITS experiment is needed. It should be highlighted that before the illumination, the irradiated HPSi samples are semi-insulating and, according to the measurements of the temperature dependence of dark current, the Fermi level is located at an energy ranging from 450 to 480 meV with respect to the conduction band minimum. The (2−/−) charge state change arises when a negatively ionized defect captures an electron from the conduction band during illumination, and while the optical pulse is switched off this electron is thermally emitted to the conduction band at the same temperature. Therefore, the concentration values given in Figures 5 and 6 for $V_2$ (2−/−), $V_3$ (2−/−), $V_4$ (2−/−), and $V_5$ (2−/−) are actually the concentrations of electrons filling the vacancy aggregates in the singly, negatively ionized state. Similarly, the concentrations values shown for $V_2$ (−/0), $V_3$ (−/0), $V_4$ (−/0), and $V_5$ (−/0) are the concentrations of electrons filling the neutral vacancies. Assuming that all the singly ionized, as well as the neutral, vacancy clusters are filled with electrons at the moment when the light is switched off, the values presented in Figures 5 and 6 may be considered as reflection of the changes in the concentration of the negatively ionized and neutral defects with an increase in the fluence. In addition, filling the negatively ionized vacancy clusters with electrons and the subsequent thermal emission occurs at lower temperatures than

the filling of the neutral clusters and their subsequent thermal emission of electrons to the conduction band.

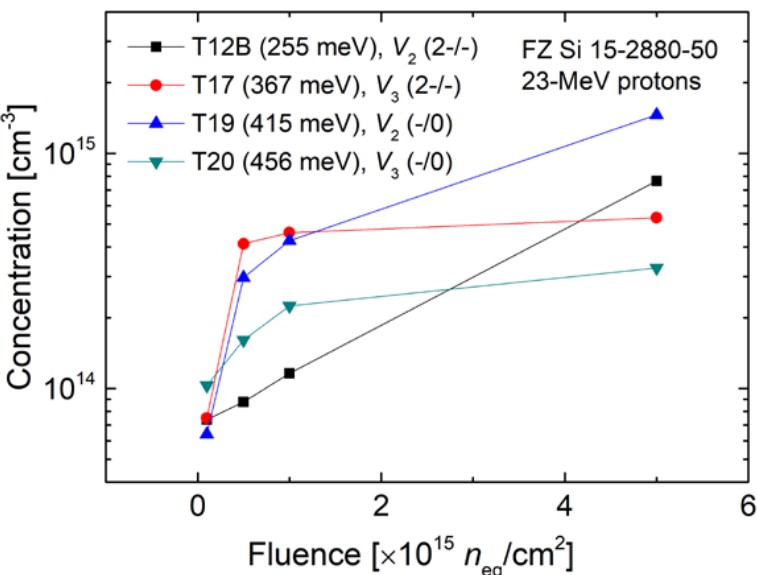

**Figure 5.** Variation in the concentrations of the T12B and T19 traps, attributed to the $(2-/-)$ and $(-/0)$ charge state changes of divacancies, respectively, as well as of the T17 and T20 traps, attributed to the $(2-/-)$ and $(-/0)$ charge state changes of trivacancies, respectively, with an increase in the fluence from $1 \times 10^{14}$ to $5 \times 10^{15}$ $n_{eq}/cm^2$ during the irradiation of HPSi samples with 23-MeV protons.

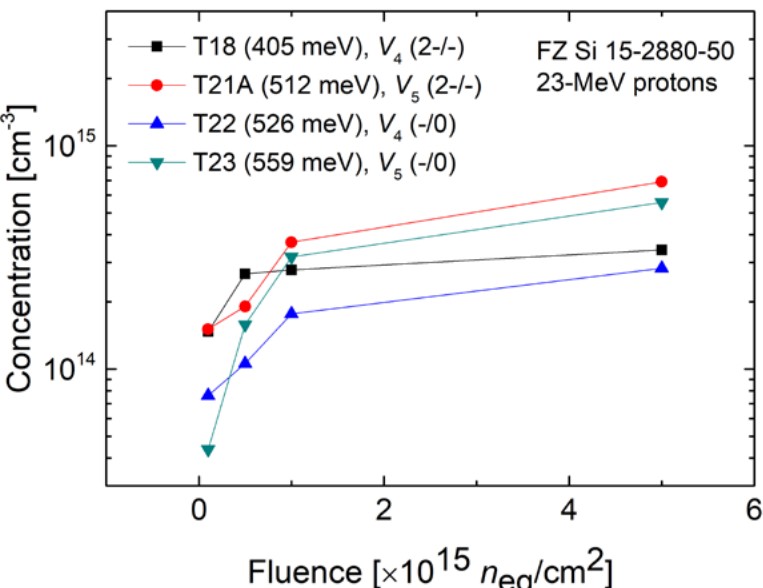

**Figure 6.** Variation in the concentrations of the T18 and T22 traps, attributed to the $(2-/-)$ and $(-/0)$ charge state changes of tetravacancies, respectively, as well as those of the T21A and T23 traps, attributed to the $(2-/-)$ and $(-/0)$ charge state changes of pentavacancies, respectively, with an increase in the fluence from $1 \times 10^{14}$ to $5 \times 10^{15}$ $n_{eq}/cm^2$ during the irradiation of HPSi samples with 23-MeV protons.

The results shown in Figures 5 and 6 indicate that for a given fluence, the concentration of each negatively ionized multi-vacancy defect may be nearly the same or different than that of the neutral one. In the case of a divacancy, for the lowest fluence, $1 \times 10^{14}$ $n_{eq}/cm^2$, the concentrations of the ionized and neutral defects, denoted as $[V_2 (-)]$ and $[V_2 (0)]$, respectively, are approximately the same. This fact allows the assumption that during the

measurements of photocurrent relaxation waveforms carried out at different temperatures, all divacancies are singly ionized or neutral and are fully filled with the excess electrons. Thus, according to the presented results (Figure 5), the divacancy concentration induced by the irradiation with this fluence is ~7 × 10$^{13}$ cm$^{-3}$. For the higher fluences, however, the $[V_2 (0)]$ values are significantly higher than those of $[V_2 (-)]$. This is because of a substantial decrease in the charge carrier lifetime induced by increasing the fluence, resulting in an insufficient excess electron concentration to make the $V_2 (2-/-)$ level, located at 255 meV below the conduction band minimum, fully occupied with electrons at the moment of the optical pulse termination [23]. Even though the excess electron concentration is lower, at the same illumination conditions the quasi-Fermi level is still located sufficiently above the $V_2 (-/0)$ level, at $E_c - 415$ meV, to introduce all divacancies into the singly, negatively ionized charge state. Therefore, for the fluences of 5 × 10$^{14}$, 1 × 10$^{15}$, and 5 × 10$^{15}$ $n_{eq}$/cm$^2$, the irradiation-induced divacancy concentrations are reflected by the $[V_2 (0)]$ values, which are equal to 3 × 10$^{14}$, 4 × 10$^{14}$, and 1.5 × 10$^{15}$ cm$^{-3}$, respectively. For the fluence of 1×10$^{14}$ $n_{eq}$/cm$^2$, the concentration of neutral trivacancy, $[V_3 (0)]$, is 1 × 10$^{14}$ cm$^{-3}$. Assuming that the equality between the neutral and negative trivacancies concentrations can be determined with the within the 50% accuracy, the $[V_3 (0)]$ value is considered to be comparable with that of negatively ionized $[V_3 (-)]$, at 7.5 × 10$^{13}$ cm$^{-3}$. Therefore, the trivacancy concentration induced by this fluence is likely to be represented by the average value of $[V_3 (0)]$ and $[V_3 (-)]$, equal to ~8.5 × 10$^{13}$ cm$^{-3}$. In the samples irradiated with the higher fluences, namely 5 × 10$^{14}$, 1 × 10$^{15}$, and 5 × 10$^{15}$ $n_{eq}$/cm$^2$, the $[V_3 (-)]$ is found to be greater than the $[V_3 (0)]$. This is due to the fact that the energy level for $V_3 (-/0)$ is located closer to the middle of the bandgap than that for $V_3 (2-/-)$. As a result of the deeper position, the excess holes from the valence band may also be captured by this level, diminishing the concentration of electrons occupying the level at the moment of switching off the optical excitation pulse, due to charge carrier recombination. In view of this fact, the trivacancy concentrations produced by these fluences are represented by the $[V_3 (-)]$ values, which are 4 × 10$^{14}$, 4.6 × 10$^{14}$, and 5.3 × 10$^{14}$ cm$^{-3}$, respectively.

In the case of tetravacancies (Figure 6), the $[V_4 (-)]$ values clearly exceed the $[V_4 (0)]$ ones, as the $V_4 (-/0)$ level, located at $E_c - 526$ meV, is nearly in the bandgap middle, and the capture of the excess holes from the valence band following the capture of the excess electrons from the conduction band is very probable. On the other hand, the $V_4 (2-/-)$ level, located at $E_c - 405$ meV, is sufficiently above the bandgap middle to make hole capture likely. Therefore, the tetravacancy concentrations induced by all fluences are represented by the $[V_4 (-)]$ values, which are 1.5 × 10$^{14}$, 2.7 × 10$^{14}$, 2.8 × 10$^{14}$, and 3.4 × 10$^{14}$ cm$^{-3}$, respectively. As far as the pentavacancy concentration is concerned (Figure 6), the $[V_5 (-)]$ values are significantly higher than the $[V_5 (0)]$ values only at the lowest fluence. This fact indicates that in the samples irradiated with the fluence of 1 × 10$^{14}$ $n_{eq}$/cm$^2$, the $V_5 (2-/-)$ level, located at $E_c - 512$ meV, is much more efficiently filled with excess electrons than the $V_5 (-/0)$ level, located in the middle of the bandgap at $E_c - 559$ meV. In the latter case, the hole capture can be strong, in particular when the generation rate of excess charge carrier is sufficiently high. In the samples irradiated with the higher fluences, in which the lifetime of charge carriers is shorter, the $[V_5 (-)]$ values are only slightly greater than that of $[V_5 (0)]$. This fact indicates that the hole capture rate is strongly diminished, although it affects the concentration of excess electrons occupying the $V_5 (-/0)$ level at the moment of switching off the optical excitation pulse. Thus, it can be assumed that the pentavacancy concentrations induced by the proton irradiations with the fluences of 1 × 10$^{14}$, 5 × 10$^{14}$, 1 × 10$^{15}$, and 5 × 10$^{15}$ $n_{eq}$/cm$^2$ are 1.5 × 10$^{14}$, 1.9 × 10$^{14}$, 3.7 × 10$^{14}$, and 6.9 × 10$^{14}$ cm$^{-3}$, respectively.

## 4. Discussion

The presented experiment and its results are unique in two key aspects. Firstly, the *n*-type HPSi samples subjected to proton irradiation had the properties that enabled vacancy cluster formation to be observed. In the samples, the concentrations of phosphorus, used as

a dopant, as well as the residual concentrations of commonly present impurities in silicon crystals, such as oxygen and carbon, were ~$2.4 \times 10^{12}$ cm$^{-3}$, ~$5 \times 10^{15}$ cm$^{-3}$, and less than ~$5 \times 10^{15}$ cm$^{-3}$, respectively. In this way, the masking effect by the formation of the phosphorus-vacancies, oxygen-vacancies, and carbon–oxygen complexes was avoided [1,2,14,24]. Secondly, the demonstrated study is the first experimental attempt to determine the energy levels of multi-vacancy defects, such as $V_2$, $V_3$, $V_4$, and $V_5$, arising at high proton fluences, ranging from $1 \times 10^{14}$ to $1 \times 10^{15}$ $n_{eq}$/cm$^2$. The objective was achieved by using the HRPITS method to investigate the defect levels in the proton-irradiated samples with a resistivity of ~150 kΩcm. In a previously reported study, the DLTS measurements were performed using the *n*-type HPSi samples exposed to proton irradiation with a fluence of $1 \times 10^{11}$ $n_{eq}$/cm$^2$ [2]. The samples resistivity before proton irradiation was ~2 kΩcm and for this fluence no substantial increase in resistivity was observed, which allowed for the DLTS technique to be applied. The energy levels at $E_c - 0.22$ eV and $E_c - 0.40$ eV, extracted from the capacitance relaxation waveforms, were assigned to $V_2$ (2−/−) and $V_2$ (−/0), respectively, but two deeper levels at $E_c - 0.35$ eV and $E_c - 0.45$ eV were proposed to be assigned to $V_3$ (2−/−) and $V_3$ (−/0), respectively [2]. Moreover, an additional level at $E_c - 0.37$ eV, which was suggested to be attributed to four-vacancy cluster ($V_4$), was found. According to the results listed in Table 1, this level could be assigned to $V_4$ (2−/−). It is worth stressing that the multi-vacancy defects, $V_4$ and $V_5$, in the neutron irradiated high resistivity silicon were identified in the EPR spectra as the Si-*P*3 and Si-*P*1 centers, respectively, several decades ago [8,18]. It has been suggested that these defects are formed in the primary cascades induced by fast neutrons, and are mainly associated with the displacement spikes which terminate the cascades [14,24].

As a result of simulation studies of Non-Ionizing Energy Loss (NIEL) in silicon exposed to various types of hadron irradiation, a model of migration and clustering of the Frenkel pairs produced by a Primary Knock-on Atom (PKA) has been developed [1]. In this model, the NIEL scaling hypothesis of silicon radiation damage was used to predict the concentrations of various defects, including $[V_2]$ and $[V_3]$, as a function of hadron fluence ranging from $1 \times 10^{12}$ to $1 \times 10^{15}$ $n_{eq}$/cm$^2$. The initial vacancy distributions were calculated for the interactions of 10-MeV and 24-GeV/c protons, as well as of 1-MeV neutrons, with the silicon crystal lattice, assuming that the displacement threshold energy $E_D$ = 20 eV. The transport of large vacancy and interstitial densities produced due to the displacement of silicon atoms from the substitutional positions was analyzed using a modified version of the TRIM (The Stopping and Range of Ions in Matter) code [1].

The simulation results indicate that the interactions involving fast neutrons tend to form isolated dense vacancy clusters surrounded by regions with low density of small vacancy clusters, small Si interstitial clusters, and vacancy–oxygen (V–O) complexes [1,14,23]. On the other hand, the irradiation with 10-MeV protons produced much smaller number of highly dense vacancy clusters, but the density of small primary defect aggregates and V–O complexes is substantially higher, and these defects are rather uniformly distributed. The 24-GeV/c protons form damaged regions also consisting of a certain number of highly dense vacancy clusters, but the density of small aggregates of primary defects and V–O complexes is several times larger than that produced by 10-MeV protons, and the spatial distribution of these defects is fairly uniform [1,14,24]. Figure 7 illustrates the $[V_2]$ and $[V_3]$ values derived from simulations based on NIEL and TRIM calculations [1] compared with those obtained from the HRPITS measurements. The simulated data are for the fluences ranging from $1 \times 10^{12}$ to $1 \times 10^{15}$ $n_{eq}$/cm$^2$ and the experimental $[V_2]$ and $[V_3]$ values are for the fluences of $1 \times 10^{14}$, $5 \times 10^{14}$, $1 \times 10^{15}$, and $5 \times 10^{15}$ $n_{eq}$/cm$^2$. It is worth stressing that fluences above $1 \times 10^{15}$ $n_{eq}$/cm$^2$ have not been considered in the simulations due to the uncertainty in the higher order multi-vacancy ($V_4$, $V_5$, and $V_6$) generation probability [1]. The presented results indicate that the experimental data closely match those resulting from the simulations up to a fluence of $1 \times 10^{15}$ $n_{eq}$/cm$^2$. This consistency enables both the experimental and simulated $[V_2]$ and $[V_3]$ values to be used to determine the trend lines illustrating the dependences of the $V_2$ and $V_3$ concentrations induced by the proton irradiation as a function of the fluence in the wide range of $1 \times 10^{12}$–$1 \times 10^{15}$ $n_{eq}$/cm$^2$. The

slopes of the $[V_2]$ and $[V_3]$ fluence dependences are $0.95 \pm 0.03$ and $0.97 \pm 0.03$, respectively. The values of $R^2$ for the fit lines depicting these dependences are equal to 0.992 and 0.995, respectively, and they unambiguously indicate that the divacancy and trivacancy concentrations linearly increase under proton irradiation with a rise in the fluence of up to $1 \times 10^{15}$ $n_{eq}/cm^2$. This finding is of great importance since it represents the first experimental confirmation that the concentrations of $V_2$ and $V_3$ formed on proton irradiation can be strongly predicted up to a fluence of $1 \times 10^{15}$ $n_{eq}/cm^2$ by the calculations based on NIEL scaling.

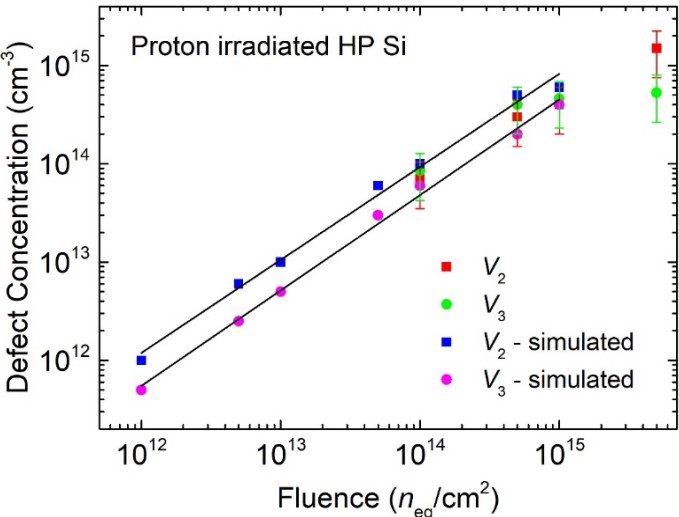

**Figure 7.** The concentrations of $V_2$ and $V_3$ formed under irradiation of HPSi samples with 23-MeV protons as a function of the fluence rising from $1 \times 10^{12}$ to $5 \times 10^{15}$ $n_{eq}/cm^2$. For the fluence up to $5 \times 10^{13}$ $n_{eq}/cm^2$, only the $[V_2]$ and $[V_3]$ simulated values obtained by NIEL calculations are taken [1]. For the fluence ranging from $1 \times 10^{14}$ to $1 \times 10^{15}$ $n_{eq}/cm^2$, both the simulated and experimental values of $[V_2]$ and $[V_3]$ are used to determine the trend lines by linear regression. For the fluence of $5 \times 10^{15}$ $n_{eq}/cm^2$, only the $[V_2]$ and $[V_3]$ experimental values are presented.

The experimental values of the $V_2$ and $V_3$ concentrations seen in Figure 7 at the fluence of $5 \times 10^{15}$ $n_{eq}/cm^2$ are lower than those expected from the trend lines. In the case of $V_2$, the difference is ~$1.5 \times 10^{15}$ $cm^{-3}$, and the values of $[V_3]$ clearly saturate at the fluence of $5 \times 10^{15}$ $n_{eq}/cm^2$ giving a difference of ~$1.0 \times 10^{15}$ $cm^{-3}$. This fact gives new insight into the formation mechanism of higher order multi-vacancies, namely $V_4$, $V_5$, and $V_6$, indicating that $V_2$ and $V_3$ play a key role in the creation of the latter, as well as contribute to creation of the disordered regions containing clusters consisting of thousands of vacancies with a size of around 100 nm [24–26]. The results of simulations based on Molecular Dynamics (MD) show that under hadron irradiation, divacancies arise due to the $V + V$ reactions and trivacancies result from the $V + V_2$ reactions [1,3]. It should be noted that according to the simulated radial distributions of the vacancy and divacancy concentrations produced by neutron-induced collisional cascades, at a distance of 2 nm from the displacement spike which terminates the cascade, the former is ~$1 \times 10^{21}$ and the latter is ~$1 \times 10^{20}$ $cm^{-3}$ [14]. At the distance of 10 nm, both concentrations are equal, at ~$1 \times 10^{19}$ $cm^{-3}$. This result supports the view that small vacancy clusters arise from the vacancies, diffusing outward from the dense region where they were created [5,14,24]. In other words, it can be assumed that $V_2$ creation results in a diminishing of the initial vacancy concentration, as well as creation of $V_3$, $V_4$, and $V_5$ diminishing some part of the $[V_2]$, $[V_3]$, and $[V_4]$, respectively. It is worth adding that at the hadron fluences above $1 \times 10^{15}$ $n_{eq}/cm^2$, the formation of the higher order multi-vacancies and disordered regions is strongly enhanced [1,5,25,26].

Figure 8 shows the $V_4$ and $V_5$ concentrations as a function of the fluence ranging from $1 \times 10^{14}$ to $5 \times 10^{15}$ $n_{eq}/cm^2$. The solid straight lines are the trend lines illustrating the rise in $[V_4]$ and $[V_5]$ with increasing the fluence of the proton irradiation and the slopes of

these lines obtained by linear regression are $0.2 \pm 0.05$ and $0.41 \pm 0.08$, respectively. The $R^2$ values for the $[V_4]$ and $[V_5]$ dependences are 0.882 and 0.880, respectively.

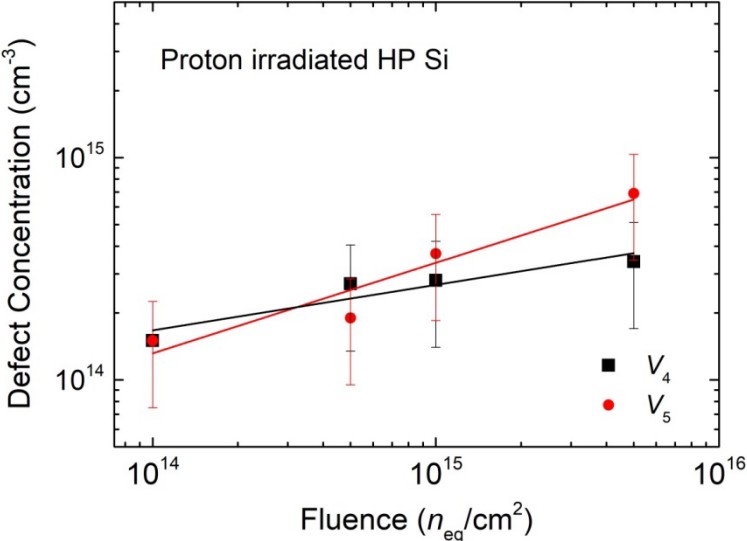

**Figure 8.** The concentrations of $V_4$ and $V_5$ formed under the irradiations of HPSi samples with 23-MeV protons as a function of the fluence rising from $1 \times 10^{14}$ to $5 \times 10^{15}$ $n_{eq}/cm^2$. The solid lines, fitted to the experimental data by linear regression, are the trend lines showing the character of the $[V_4]$ and $[V_5]$ changes with the fluence.

An important feature of the presented results is that the lines cross at the fluence of about $1 \times 10^{15}$ $n_{eq}/cm^2$. This fact means that for the fluences approximately below $1 \times 10^{15}$ $n_{eq}/cm^2$, the concentration of pentavacancies produced under the proton irradiation is lower than that of tetravacancies, but for the fluences approaching the value of $1 \times 10^{15}$ $n_{eq}/cm^2$ or exceeding it, the $[V_5]$ becomes greater than the $[V_4]$. Using the results illustrated in Figure 8, we can formulate the following empirical formulas describing the fluence dependences of $V_4$ and $V_5$ in the proton-irradiated HPSi samples.

$$[V_4] \approx 10^{11.4} \times \Phi^{0.2} \, [cm^{-3}], \tag{6}$$

$$[V_5] \approx 10^{8.4} \times \Phi^{0.41} \, [cm^{-3}]. \tag{7}$$

According to these formulas, at the fluences of $1 \times 10^{13}$ and $1 \times 10^{16}$ $n_{eq}/cm^2$, the $[V_4]$ values are $1 \times 10^{14}$ and $4 \times 10^{14}$ $cm^{-3}$, respectively, and the $[V_5]$ values are $5.4 \times 10^{13}$ and $9.1 \times 10^{14}$ $cm^{-3}$, respectively.

A point to be stressed is that the multivacancies whose concentrations rise with the fluence, as shown in Figures 7 and 8, were produced at room temperature (~300 K). During proton irradiation, the sample's temperature was precisely controlled to minimize the heating involved with increasing the fluence. At 300 K, the mobility of primary defects, namely monovacancies and single Si interstitials ($I$), is substantially higher than that of complex defects resulting from their interactions [1,3,14]. Therefore, the main factor determining the concentrations of various types of multivacancies produced in the material at a given fluence is the monovacancy concentration induced by this fluence. However, the $V$ density changes during the time of the irradiation and at the end of the process is too small to be measured. The decrease in the $V$ concentration is mainly due to the recombination with the Si interstitials. The probability of the $V + I \rightarrow Si$ reaction, determined with respect to the predefined capture radius of 1.62 nm corresponding to the three Si lattice constants, is 0.956 [1]. The vast majority of vacancies remain after the recombination and contribute to the formation of $V_2$. Although the $V + V \rightarrow V_2$ reaction probability is 0.107 and is more than two times lower than that of $V_3$ creation, which according to the reaction $V + V_2 \rightarrow V_3$ is 0.226 [1], the $V_2$ concentration seen in Figure 7 is clearly above the concentration of $V_3$.

This phenomenon has been predicted by simulations [1] and is confirmed by the HRPITS results (Figure 7) because at the time when $V_2$ are created, the monovacancy concentration is significantly higher than when $V_3$ are formed [1,3,14]. In other words, the multi-vacancy defects are formed in a sequence of events; the lower order defect must be created before the higher order one can arise. Since the size of the latter is larger, the probability of its capture by a monovacancy is higher. On the other hand, however, the concentration of monovacancies yet to be captured by a higher order multivacancy is lower [14,24].

## 5. Conclusions

The unique experimental studies aimed at determining the energy levels related to the $(2-/-)$ and $(-/0)$ charge state changes of multi-vacancy defects in high-purity silicon have been performed. To find the properties and concentrations of small vacancy clusters, namely $V_2$, $V_3$, $V_4$, and $V_5$ produced in the material by the 23-MeV proton irradiations, the state-of-the-art HRPITS technique has been used. A distinguished feature of this technique is the implementation of a numerical procedure based on the inverse Laplace transformation algorithm to the analysis of the photocurrent relaxation waveforms. This is used to extract the temperature dependence of the emission rate of the excess charge carriers released from deep-level defects involving the changes in their charge states. The motivation behind the studies was to experimentally verify theoretical predictions concerning the energy levels related to the charge state changes of small vacancy clusters such as $V_2$, $V_3$, $V_4$, and $V_5$, as well as to establish the effect of fluence ranging from $1 \times 10^{14}$ to $5 \times 10^{15}$ $n_{eq}/cm^2$ on their concentration in the material at room temperature.

It is postulated that the energy levels associated with $V_2$ $(2-/-)$, $V_3$ $(2-/-)$, $V_4$ $(2-/-)$, and $V_5$ $(2-/-)$ are located in the Si bandgap at the energies of 255, 367, 405, and 512 meV relative to the conduction band minimum, respectively. In agreement with the theoretical predictions, the energy levels associated with $V_2$ $(-/0)$, $V_3$ $(-/0)$, $V_4$ $(-/0)$, and $V_5$ $(-/0)$ are at a deeper location in the Si bandgap at the energies of 415, 456, 526, and 559 meV relative to the conduction band minimum, respectively. These results are consistent with the previously reported energy levels for $V_2$ $(2-/-)$ and $V_2$ $(-/0)$ as well as for $V_3$ $(2-/-)$ and $V_3$ $(-/0)$ found by DLTS measurements.

It is also argued that the $V_2$ and $V_3$ concentrations introduced by proton irradiation linearly increase with the fluence up to $1 \times 10^{15}$ $n_{eq}/cm^2$, and the values of these concentrations are consistent with the previously reported simulated data obtained by calculations based on NIEL scaling. The linearity is destroyed at the fluence of $5 \times 10^{15}$ $n_{eq}/cm^2$, at which the $V_4$ and $V_5$ concentrations values, equal to $3.4 \times 10^{14}$ and $6.9 \times 10^{14}$ $cm^{-3}$, respectively, become closer to that of $V_2$ and $V_3$, which are $1.5 \times 10^{15}$ and $5.3 \times 10^{14}$, respectively.

The $V_4$ and $V_5$ concentrations are found to non-linearly increase with the fluence in the whole range of $1 \times 10^{14}$–$5 \times 10^{15}$ $n_{eq}/cm^2$. The empirical equations describing these multi-vacancy defect concentrations as a function of the fluence have been proposed. It is argued that the $V_2$ and $V_3$ clusters directly and/or indirectly participate in $V_4$ and $V_5$ defect formation, and at a sufficiently high fluence the rise in concentration of the latter can involve a fall in the concentration of the former.

The presented results show the possibility to experimentally investigate the electronic properties of multi-vacancy defects formed at the hadron fluences of $5 \times 10^{15}$ $n_{eq}/cm^2$ or higher. Deeper understanding of the primary defect clustering may lead to finding a way of improving the radiation hardness of silicon and other materials. The investigations in this direction have already been undertaken by a number of research groups. The radiation damage produced by high fluences of hadrons still remains an area to be explored. Under hadron irradiations with these fluences, silicon, as well as other semiconductor materials, become semi-insulating. HRPITS seems to be a powerful technique that could be used for further studies of the properties and concentrations of point defect clusters either in as-irradiated or annealed samples. Moreover, the deep-level defects in semi-insulating samples can be investigated by both the HRPITS and EPR methods, and the results obtained

by the combination of these experimental techniques would allow verification of many theoretical predictions and greatly enrich the knowledge on radiation damage.

**Author Contributions:** Conceptualization, P.K.; methodology, P.K., R.K., C.H. and L.J.; software, J.Ż.; validation, P.K. and C.H.; investigation, P.K. and R.K.; resources, C.H. and L.J.; data curation, J.Ż.; writing—original draft preparation, P.K.; writing—review and editing, R.K. and J.Ż.; visualization, J.Ż. and R.K.; supervision, P.K. and C.H.; project administration, P.K.; funding acquisition, P.K. and C.H. All authors have read and agreed to the published version of the manuscript.

**Funding:** This research was financially supported by the Topsil Semiconductor Materials A/S, Denmark, as well as by the National Centre for Research and Development in Poland, within the framework of grant number PBS2/A9/26/2014 dedicated to the NitroSil project (ID: 208346) in the Applied Research Program.

**Data Availability Statement:** The data presented in this study are available on request from the corresponding author.

**Acknowledgments:** We would like to thank our colleague Barbara Surma for determining the oxygen and carbon concentrations in HPSi samples by FTIR measurements. We also wish to express our appreciation to Alexander Dierlamm (Karlsruhe Institute of Technology) for his aid in the proton irradiations by means of the state-of-the-art facilities. Special thanks from P.K., J.Ż., and R.K. are due to Gunnar Lindström (Hamburg University) for his stimulating activity within the RD50 Collaboration and beneficial advice on pursuing the studies of radiation damage in semiconductor materials, as well as due to Michael Moll (CERN) and Eckhart Fretwurst (Hamburg University) for valuable discussions and suggestions.

**Conflicts of Interest:** The authors declare no conflict of interest. The funders had no role in the design of the study; in the collection, analyses, or interpretation of data; in the writing of the manuscript; or in the decision to publish the results.

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
