# Peer review of "Investigation of Energy Levels of Small Vacancy Clusters in Proton Irradiated Silicon by Laplace Photoinduced Transient Spectroscopy"

_crystals, doi:10.3390/cryst12121703_

Round 1
Reviewer 1 Report
The manuscript under consideration presents experimental results and their interpretation coming from a high-purity n-type silicon sample irradiated with protons. The formation of vacancy clusters is carefully investigated using Laplace photoinduced transient spectroscopy. The topic is of high interest in nuclear and particle physics because Si-detectors are the workhorse of particle detection in many fields and detector damage after irradiation has been investigated up to these fluences for decades. The results presented here are, in my opinion, quite interesting because of the focus on multi-vacancies formation and dynamics, which are still poorly understood for fluences higher than 10^10 neq/cm2.
The experimental and analyses techniques are solid and properly explained in the manuscript. Concerning the theoretical interpretation, I tend to agree with most of the conclusions drawn by the authors, with a few exceptions detailed below. The paper is reasonably well written even if I suggest a language revision. To help the authors, I added a few language corrections at the end of my report. The manuscript is clear and detailed. The discussion and conclusion sessions could be shortened since much information is repeated twice in the paper but length is not a major issue and I leave the author (and editor) to decide whether it is worth shortening sec. 4 or keeping it in its present form.
In conclusion, I believe the manuscript fits the acceptance criteria of Crystals and I support its publication after the authors address the comments below.
Comments:
1) Lines 120-128 (p. 3) Non-negligible concentrations of elements different from C and O may mimic vacancies and explain some of the discrepancies noted in the paper. Are you able to claim that the only contaminations of relevance for your study are C and O?
2) Line 139 p.3 100 hours is an unusually long time for preparation and may affect the dynamics of vacancies but this potential source of systematic uncertainty is not discussed in the paper. Can you comment on it and/or extend the discussion of the potential effect arising from the post-irradiation sample preparation?
3) Line 174 It is a good practice to use photon flashes of the same intensity (if possible). I see you used two photon fluxes for low and high-irradiation samples. Why?
4) Lines 220-223 Why I(0) should be (linearly) proportional to I_L I_ph? I see that this is somehow an empirical relation you used to estimate I(0) but a non-linear effect can bias your estimation. Can you comment on it?
5) Fig. 1 and 2 Please replace “1E14” with “10^14”
6) Lines 358 and 403. How did you estimate the 5% experimental uncertainty? Instrumental uncertainties were not discussed in the manuscript and it is important to provide readers with more details
7) Lines 529-531(p.13) I don’t think you have an experimental error of 50%. Maybe a better statement would be to say that the equality between neutral and negative trivacancies is valid within 50%.
8) Figure 7 I cannot see error bars below 10^14 n/cm2. Are these errors estimated or are they smaller than the markers in the plot?
9) Line 647-651 It is hard to make any inference on the role of V2 if we are not aware of the uncertainty associated with the fit line. Can you really claim that the discrepancy between the fit and the V2 values can justify the role of V2 and V3 in the creation of V4 and V5?
Language editing (selection)
l. 52 remove “as-irradiated”
l.60 remove “particular” and “:”
l.61 “predicted” -> “are predicted”
l.62 “, are” -> “to be”
l.101 remove “in n-type”
l.105 “the state-of-the-art” -> “state-of-the-art” (remove “the”)
l. 106 “the high-resolution” -> “high-resolution” (remove “the”)
l.110 “similarly as in the case of” -> “, like”
l.111 “the higher” remove “the”
l.117 remove “as-grown”
l.179 “with an increment” -> “in steps”
l.217 “at the moment of” -> “when”
l.314 “we can tentatively assign” -> “can be tentatively assigned”
l.317 “favor” -> “favored”
l. 359 “match perfectly giving the” -> “perfectly match and provide”
l.384 “demonstrated” -> “shown”
l.395 remove “presented”
l.470 “the both figures” -> “these figures”
l.476 “to arising” -> “to the formation of”
l.573 “to investigating” -> “to investigate”
Author Response
The authors are very grateful to the Reviewer for his valuable questions and comments. Below are the answers to the questions and comments and the changes introduced into the manuscript.

Reviewer 2 Report
The authors need to make minor changes to the text of the article:
In line 363, replace the designation of the test sample T17 with T18;
In line 368, replace the designation of the test sample 20 with T20;
Make corrections to the sentence on lines 700 - 702
Author Response
Response to comments of Reviewer 2
The authors are very grateful to the Reviewer for his comments. All the Reviewer’s suggestions were taken into account in the revised manuscript.
Comment 1
In line 363, replace the designation of the test sample T17 with T18;
Answer
The correction has been made in the manuscript.
Comment 2
In line 368, replace the designation of the test sample 20 with T20;
Answer
The correction has been made in the manuscript.
Comment 3
Make corrections to the sentence on lines 700 – 702
Answer
The corrections has been made in the manuscript.
On behalf of the authors
Jarosław Żelazko